

# Frequency and developmental timing of linear enamel hypoplasia defects in Early Archaic Texan hunter-gatherers

J. Colette Berbesque[1] and Kara C. Hoover[2]

[1] Centre for Research in Evolutionary, Social and Inter-Disciplinary Anthropology, University of Roehampton, London, United Kingdom

[2] Department of Anthropology, University of Alaska Fairbanks, Fairbanks, AK, United States of America

## ABSTRACT

Digital photographs taken under controlled conditions were used to examine the incidence of linear enamel hypoplasia defects (LEHs) in burials from the Buckeye Knoll archaeological site (41VT98 Victoria county, Texas), which spans the Early to Late Archaic Period (ca. 2,500–6,500 BP uncorrected radiocarbon). The majority (68 of 74 burials) date to the Texas Early Archaic, including one extremely early burial dated to 8,500 BP. The photogrammetric data collection method also results in an archive for Buckeye Knoll, a significant rare Archaic period collection that has been repatriated and reinterred. We analyzed the incidence and developmental timing of LEHs in permanent canines. Fifty-nine percent of permanent canines ($n = 54$) had at least one defect. There were no significant differences in LEH frequency between the maxillary and mandibular canines ($U = 640.5$, $n1 = 37$, $n2 = 43$, $p = .110$). The sample studied ($n = 92$ permanent canines) had an overall mean of 0.93 LEH defect per tooth, with a median of one defect, and a mode of zero defects. Average age at first insult was 3.92 (median = 4.00, range = 2.5–5.4) and the mean age of all insults per individual was 4.18 years old (range = 2.5–5.67). Age at first insult is consistent with onset of weaning stress—the weaning age range for hunter-gatherer societies is 1–4.5. Having an earlier age of first insult was associated with having more LEHs ($n = 54$, rho = $-0.381$, $p = 0.005$).

## INTRODUCTION

The Buckeye Knoll site (41VT98) contains a prolonged record of short-term continuous site use over a period of 8,000 years (8,500–500 BP) with evidence of resource caching for future occupations. We know very little about Archaic life history and Buckeye Knoll constitutes one of the largest populations available for testing hypotheses regarding health and disease in this early period of North American prehistory. Excavation uncovered 75 discrete burial loci and recovered a minimum number of 116 individuals that were dated to 8,500–3,500 BP using tooth and bone collagen samples. Buckeye Knoll was exhumed and reburied in compliance with the Native American Graves Protection and Repatriation Act (NAGPRA), so any future data collection or analysis must come from the digital photographs collected for archival purposes (*Ricklis, Weinstein & Wells, 2012c*).

Corresponding author
J. Colette Berbesque,
colette.berbesque@roehampton.ac.uk

Dental enamel hypoplasia defects represent an interruption in the growth process of teeth and can be attributed to genetics (*Brook, 2009*; *Hart et al., 2002*; *Zilberman et al., 2004*), trauma (*Brook, 2009*), and insult (*Goodman, 1988*; *Sarnat & Schour, 1942*; *Sarnat & Schour, 1941*). Those linked to external biological insult (e.g., foreign disease pathogen, injury) develop when resources normally directed to growth and development are rerouted to defending the body or are only insufficient to sustain maintenance activities (e.g., malnourishment, diarrhea) (*Sarnat & Schour, 1942*; *Sarnat & Schour, 1941*). Enamel hypoplastic defects occur on the buccal and labial surfaces of teeth and mostly commonly manifest as transverse grooves, or linear enamel hypoplasia (LEH), but also can appear as pits or grooves (*Hillson & Bond, 1997*). Because teeth do not remodel, defects captured during growth and development are permanent and have been used to infer early life health in a number of populations (e.g., *Berbesque & Doran, 2008*; *Guatelli-Steinberg, Larsen & Hutchinson, 2004*; *Hoover & Matsumura, 2008*; *Lieverse et al., 2007*; *Temple, 2010*). Of particular note are the associations between weaning stress (e.g., *Herring, Saunders & Katzenberg, 1998*; *Katzenberg, Herring & Saunders, 1996*; *Moggi-Cecchi, Pacciani & Pinto-Cisternas, 1994*) and earlier age at death (*DeWitte & Stojanowski, 2015*; *Walter & DeWitte, 2017*; *Yaussy, DeWitte & Redfern, 2016*).

A major shift in dietary pattern and environmental adaptations occurred in the southern United States during the transition from early to mid-Holocene. This period was a time of dramatic worldwide changes in temperature, sea level, and coastal 'configuration'. Buckeye Knoll may have been in a period of climatic transition, the severity of which is unknown. The climate reconstruction of Buckeye Knoll was primarily from palynology. Two cores were taken from the Guadalupe River Flood Plain adjacent to the Buckeye Knoll Site for palynological analysis. These cores enable a regional vegetation reconstruction extending back to 9,500 cal. B. P. until present. During this period, there were marked changes in climate reflected in the pollen taxa represented, particularly circa 6,000 BP when climate change resulted in enough increases in upland-prairie biomass that it may have caused a shift in subsistence strategy (*Ricklis, Weinstein & Wells, 2012a*). This might be a factor in the overall levels of systemic stress in populations of this time period, such as Buckeye Knoll. Here, we aim to infer nonspecific nutritional and developmental stresses via the developmental timing and frequency of linear enamel hypoplasia defects (LEH) in the canines using photogrammetric methods.

## METHODS

### Study site description

The first evidence for human activity at Buckeye Knoll dates to the Paleo-Indian period and consists of scattered artifacts, specifically stone darts. Prolonged occupation of the site begins in the Archaic period, which is marked by a variety of human activities linked to repeated short-term occupation. Primary artifacts include debitage, projectiles, tools, beads, bone, shell, and hearths. More recent artifacts include indigenous ceramics. The site record contains evidence for a prolonged record of short-term continuous use for a period of 8,000 years (8,500–500 BP). Of particular interest are large pits which may

have been used to store food which suggests longer occupations of up to a few months; even more interesting is evidence for material caching which suggests intentional regular re-occupation (*Ricklis, Weinstein & Wells, 2012c*).

Faunal remains recovered from the site are abundant—74,000 identifiable fragments representing a minimum of 126 vertebrate taxa including fish (mostly gar), small mammals (often rodents), some large mammals (e.g., deer), and rarely birds. The pattern of resource exploitation evidenced by faunal analysis suggests that opportunistic hunting of larger game was gradually replaced by increased emphasis on net-fishing (evidenced by a shift from larger to smaller fish body sizes) and wider exploitation of other taxa; this may be attributable to increased population demands over time (*Ricklis, Weinstein & Wells, 2012c*) or the previously noted climate change that resulted in changes to the local environment and possible dietary shifts in response to that change.

A total of 75 discrete burials containing 119 individuals were excavated. The majority of burials were single interment but there were also graves containing multiple individuals. All but one burial (dated to the Late Archaic) were interred on the Knoll Top. Of the remaining 74 burials, the vast majority ($n = 68$) date to the Texas Early Archaic, including one extremely early burial dated to 8,500 BP. The Texas Early Archaic burial dates tend to cluster between 7,400–6,300 BP–the lack of non-mortuary activity at the site during the 7th millennium (roughly 7,000–6,200 BP) suggests that the Knoll Top space was reserved exclusively for treatment of the dead during this time (*Ricklis, Weinstein & Wells, 2012b*; *Ricklis, Weinstein & Wells, 2012c*). Texas Early Archaic burials are associated with artifacts that form a unique mortuary assemblage that is closely related to Middle Archaic period (i.e., ca. 8,000–5,000 BP) cultures in the Mississippi Valley region and beyond. Thus, this assemblage reflects larger regional cultural associations. During this period, flexed or semi-flexed burials were most common followed by a smaller number of disarticulated individuals, and an even smaller number of individuals interred in sitting postures. The Late Archaic period was characterized by extended burials (*Ricklis, Weinstein & Wells, 2012b*).

## Photogrammetric materials and methods

Photographs were used for data collection because the Buckeye Knoll sample was reinterred. Reliability of LEH scoring is more robust in photogrammetric methods, with a significant increase in LEH number identified compared to direct examination method (*Golkari et al., 2011*). This method was successfully applied to a similar published study on another Early Archaic population, Windover (*Berbesque & Doran, 2008*).

Photographs were taken of the left maxillary and mandibular canines using the Nikon 990 Coolpix in macro mode. The diminished focal length presents some difficulty with depth or focus on anything other than one plane. As teeth are often curved, every attempt was made to capture the labial surface of the tooth with most clarity. Multiple photographs were taken from different angles to ensure defects were scorable. A metric scale was placed in the plane of the tooth surface in each photograph. The photographs were taken in high quality TIFF file format. Missing teeth or teeth too worn to score were excluded from analysis. In some cases, dental calculus prevented an accurate measurement of crown height, and measurements were then taken from the bottom of the calculus to the top

of the crown. These measurements are primarily for quality control in using an imaging software for analysis.

Permanent canines were chosen for data collection because they have a prolonged period of crown formation (7.5 months to 6.5 years for maxillary canines and 10.5 months to 5.5 years for mandibular canines) (*AlQahtani, Hector & Liversidge, 2014*) and can best capture the peak window of developmental stress caused by weaning (*Sandberg et al., 2014*). LEH was scored in Microsoft Paint. Once scored, the images were imported into Scion Image for analysis (a PC friendly software modeled after the National Institute of Health ImageJ, which is commonly used in morphometrics studies) (*Scion, 2000–2001*) http://www.nist.gov/lispix/imlab/labs.html.

Developmental timing of each defect was determined using the estimate by Reid and Dean (*Reid & Dean, 2000*), which necessitates estimation of complete, unworn crown height for every tooth. An estimate of completeness for each canine was based on surrounding dentition and other canines within the population. The median percent complete for permanent dentition is 85% overall. Mandibular canines were 86% complete, and maxillary canines were 81% complete. This visual estimate of complete canine height provided a wear estimate for each canine. Because this population has significant dental wear, stage of development for each defect was determined by measuring the distance from the cemento-enamel junction to the bottom of each defect rather than from the tip of the cusp down to the defect. All statistical analysis was conducted using SPSS version 22. None of the variables met the assumptions of a normal distribution, so nonparametric statistics were used for all analyses.

To place Buckeye Knoll in context with similar populations, data from this study were compared to published data from populations dating to an average of 3,000 years or older contained in the public *Global History of Human Health Database* (*Steckel & Rose, 2002*) (see Table 1). Buckeye Knoll was also compared with another Early Archaic population, Windover (8,120–6,980 14C years B.P. uncorrected), using the same methods deployed in this study (*Berbesque & Doran, 2008*).

## RESULTS

There were 41 deciduous canines in the sample and 92 permanent canines. The permanent dentition consisted of 37 maxillary canines and 43 mandibular canines—12 could not be identified as maxillary or mandibular. The permanent dentition had a hypoplasia frequency rate of 59% ($n = 54$ canines with at least one hypoplastic defect) in the population. There was an overall mean of 0.93 defects per permanent canine, with a median of one defect, and a mode of zero defects. We did not analyse deciduous dentition for timing of defects. Out of 41 deciduous canines in the population, only one defect was found.

Despite limited demographic information available for these mostly isolated dentition, there were associated skeletal material for some individuals—allowing for a basic breakdown by sex and age category (adults versus juvenile with permanent dentition). Juveniles with permanent dentition had higher rates of multiple defects than the general population (see Table 2). Table 2 provides breakdown of the sample by presenting frequency

**Table 1 Descriptive information for comparative sites, including domesticated plants/animals.**

| Site | n | Animals | Plants | Climate | Settlement | Site date |
|------|---|---------|--------|---------|------------|-----------|
| Preceramico | 60 | None | None | Subtropical | Mobile | 2,000–4,000 |
| Tlatilco | 80 | Some | Maize, beans, squash | Temperate | Small/medium village | 2,930–3,250 |
| Realto | 34 | Some | None | Tropical | Settled dispersed | 3,450–5,876 |
| Sta. Elena | 39 | None | None | Tropical | Mobile | 6,600–8,250 |
| Buckeye Knoll | 92 | None | None | Subtropical | Mobile | 3,500–8,500 |

**Table 2 LEH count and frequency by demographic category, Buckeye Knoll.**

| | Total n | 0 LEH | | 1 LEH | | 2 LEH | | 3 LEH | | 4 LEH | |
|---|---------|-------|------|-------|------|-------|------|-------|------|-------|------|
| | | n | Freq | n | Freq | n | Freq | n | Freq | n | Freq |
| Males | 5 | 1 | 0.20 | 2 | 0.40 | 1 | 0.20 | 1 | 0.20 | 0 | 0.00 |
| Females | 13 | 5 | 0.38 | 5 | 0.38 | 2 | 0.15 | 1 | 0.08 | 0 | 0.00 |
| Juveniles | 6 | 0 | 0.00 | 1 | 0.17 | 0 | 0.00 | 3 | 0.50 | 2 | 0.33 |
| Adult[a] | 9 | 7 | 0.78 | 1 | 0.11 | 1 | 0.11 | 0 | 0.00 | 0 | 0.00 |
| Canines[b] | 59 | 25 | 0.42 | 23 | 0.39 | 8 | 0.14 | 2 | 0.03 | 1 | 0.02 |

**Notes.**
[a] No sex identification.
[b] Loose, not affiliate with any burial.

and portion of the overall sample by LEH count (range = 0–4) and demographic category.

There were no significant differences between the maxilla and mandible in timing of earliest defect (Mann Whitney $U = 228$, earliest maxillary defect $N = 20$, earliest mandibular $N = 27$, $p = .366$) or number of defects ($U = 640.5$, maxillary defects $N = 37$, mandibular defects $N = 43$, $p = .110$). The mean age for the earliest defect per individual was 3.92 (range = 2.5–5.4). Individuals with more LEHs also had earlier age of first insult ($n = 54$, rho $= -0.381$, $p = 0.005$). The mean developmental age of all defects was 4.18 years old (range = 2.5–5.67).

A comparative analysis of individual LEH frequency in Buckeye Knoll and populations in the Global History of Human Health Database (*Steckel & Rose, 2002*) found that Buckeye Knoll frequencies were significantly higher with one or more LEH on their canine (see Table 3) (Chi-Square = 58.425, $df = 4$, $p = 0.000$).

LEH incidence in another Early Archaic population, Windover, was more than twice that of Buckeye Knoll (see Table 4) (*Berbesque & Doran, 2008*). LEH data collection methods for both sites used the same photographic methods.

## DISCUSSION AND CONCLUSIONS

Juveniles with permanent dentition had the highest incidence of LEH. Also, having greater numbers of LEH defects was associated with earlier age of death, providing some evidence for a mortality curve that would support the use of LEH as a stress indicator in this population and indicating social factors that warrant further investigation. This finding provides some evidence for the Barker Hypothesis; wherein individuals exposed to stressors

**Table 3  LEH count and frequency, comparative populations.**

| Site | Total $n$ | 0 LEH | | 1 LEH | | 2 + LEH | |
|---|---|---|---|---|---|---|---|
| | | Count | Freq | Count | Freq | Count | Freq |
| Preceramico | 60 | 41 | 0.68 | 16 | 0.27 | 3 | 0.05 |
| Tlatilco | 80 | 41 | 0.51 | 32 | 0.40 | 7 | 0.09 |
| Realto | 34 | 31 | 0.91 | 3 | 0.09 | 0 | 0.00 |
| Sta. Elena | 39 | 38 | 0.97 | 1 | 0.03 | 0 | 0.00 |
| Buckeye Knoll | 92 | 38 | 0.41 | 32 | 0.35 | 22 | 0.24 |

**Table 4  LEH descriptive statistics, Buckeye Knoll and Windover.**

| | Mandibular canine | | Maxillary canine | |
|---|---|---|---|---|
| | Windover | Buckeye Knoll | Windover | Buckeye Knoll |
| N | 59 | 43 | 48 | 37 |
| Mean LEH | 2.78 | 1.07 | 2 | 0.7 |
| Median LEH | 3 | 1 | 2 | 1 |
| Mode LEH | 3 | 0 | 2 | 0 |
| Range | 1–6 | 1–4 | 1–4 | 1–4 |
| SD | 1.34 | 1.06 | 0.99 | 0.85 |

earlier in life may actually have damaged immunological competence as a consequence of those stressors (*Armelagos et al., 2009*; *Goodman & Armelagos, 1989*).

The location of each defect gives insight into the timing of metabolic insult. Cusp enamel completion occurs at 1.7 years for maxillary canines and 0.98 years for mandibular canines (*Reid & Dean, 2000*). As the first period on the occlusal surface of the crown is often worn away by attrition, much of the data on the second year of life is lost. Clustering of LEH around a location on the tooth that corresponds to a particular age might indicate some stressful milestone event whether culturally flexible (e.g., age of weaning) or not (e.g., birth). Weaning ages across hunter-gatherer societies vary considerably, with New World hunter-gatherers weaning earlier (mean = 2.32 years old) than Old World hunter-gatherers (mean = 3.20 years old) and a combined range of 1 to 4.5 (*Marlowe, 2005*). Age of the mean earliest defect for Buckeye Knoll is within this range (mean = 3.92), but late for the mean age of weaning in ethnographically described hunter-gatherers in the New World. Perhaps the developmental timing of most LEH defects has less to do with extreme stress from weaning and more with the more with the acute angles formed by the Striae of Retzius relative to the enamel surface to enamel formation. It has been suggested that these acute angles make even small disruptions in enamel production are more pronounced and visible in the intermediate and occlusal thirds of the tooth (*Blakey, Leslie & Reidy, 1994*; *Newell et al., 2006*).

Of the limited samples of comparable antiquity (minimally over 3,000 years old on average) in the Global History of Human Health Database (*Steckel & Rose, 2002*; *Steckel, Sciulli & Rose, 2002*), most populations demonstrated lower incidence of LEH compared to Buckeye Knoll (59% with at least one defect). The comparative sample with the closest

frequency of Buckeye Knoll LEH was Tlatilco. Tlatilco was a sedentary population with evidence of domesticated plants and animals. Sedentary populations and those using domesticated plants were found to have higher incidence of various stress indicators, and agriculturalists are documented as having higher LEH incidence than foragers (*Larsen, 1995*; *Starling & Stock, 2007*).

It has been suggested that fishing populations might be at higher risk for LEH defects due to parasite load (*Bathurst, 2005*). One example of this is found in Japan; prehistoric hunter-gatherer-fishers have surprisingly high rates of LEH but these are sedentary complex stratified populations (*Hoover & Matsumura, 2008*; *Temple, 2010*). And, the higher incidence of defects is widely documented across the island and throughout time; given the abundance of resources and consistently high rates of LEH, a likelier explanation might be a genetic etiology (*Hoover & Hudson, 2016*; *Hoover & Matsumura, 2008*; *Hoover & Williams, 2016*). Coastal populations share a host of traits that may contribute to LEH defects, such as sedentism and reliance on domesticates. Although the Buckeye Knoll population likely relied at least partially on coastal resources, there is no evidence of domesticated plants or animals or sedentism at Buckeye Knoll.

The population most comparable to Buckeye Knoll is Windover. Windover has been assessed for LEH defects using the same methods used in the GHHD as well as the photogrammatic methods. Even when examining data on LEH defects using the unaided eye, Windover had a very high number of individuals affected by LEH defects. In the GHHD, 100% represents a population completely unaffected by LEH, and the GHHD score for LEH in Windover was = 39.5% (*Wentz et al., 2006*). It is not clear why these two Early Archaic populations both appear to have a surprisingly high incidence of LEH, but a possible ecological explanation for the high overall incidence of LEH defects in this population is the climate shift during this time that may have caused physiological stress during periods of diminished resources.

## CONCLUSIONS

Buckeye Knoll had greater incidence of LEH than any other population in the Global History of Health Database of comparable age. However, these data are taken by unaided visual assessment only, and photogrammetric methods have been shown to result in identification of greater numbers of LEH defects. However, Buckeye Knoll had fewer LEH defects compared with data collected using the same photogrammetric methods from Windover, a population of comparable antiquity. It is not clear whether the higher incidence of defects seen in these populations are entirely due to methodological differences in data collection, or whether an environmental factor such as the climate change documented during the Early Archaic period affected the health of coastal/riverine foragers such as the Windover and Buckeye Knoll populations.

## ACKNOWLEDGEMENTS

We thank Dr. Glen Doran for access to the Buckeye Knoll population.

### Funding

The US Army Corps of Engineers, Galveston District, commissioned the excavation and repatriation of this population. The funders had no role in study design, data collection and analysis, decision to publish, or preparation of the manuscript.

### Grant Disclosures

The following grant information was disclosed by the authors:
The US Army Corps of Engineers.

### Competing Interests

Kara C. Hoover is an Academic Editor for PeerJ.

### Author Contributions

- J. Colette Berbesque conceived and designed the experiments, performed the experiments, analyzed the data, wrote the paper, prepared figures and/or tables, reviewed drafts of the paper.
- Kara C. Hoover wrote the paper, prepared figures and/or tables, reviewed drafts of the paper.

### Data Availability

The raw data on Buckeye Knoll are included with data on developmental timing of defects. Also, the recoded Buckeye Knoll data (coding system and data available here: Available at http://global.sbs.ohio-state.edu/western_hemisphere_module.htm) has been inserted into the GHHD dataset for comparison.

### Supplemental Information

Supplemental information for this article can be found online at http://dx.doi.org/10.7717/peerj.4367#supplemental-information.

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
