# Peer review of "Frequency and developmental timing of linear enamel hypoplasia defects in Early Archaic Texan hunter-gatherers"

_PeerJ, doi:10.7717/peerj.4367_

## Round 0.1 · original submission · Major Revisions

The authors are invited to address the issues raised, particularly in the review from the 1st reviewer, and resubmit.

Reviewer 1 ·

Basic reporting

- The paper is well written and in this context it do not need any changes.
- The literature is correctly selected, although you can delete the oldest quotes (e.g., Sarnat 1941, 1942).
- The structure of manuscript is very confused.In paragraph "Methods" the authors write about materials. It is not known why they gave an additional sub-section "Study population" (line 72). In the subsection "materials" they described using methods (line 103).
-some tables are illegible (e table 2) - a column with a number of teeth was introduced, but it is not clear why there are so many rows.

Experimental design

-The aim of the study was not clear stated.
-The teeth with strong wear were excluded (line 115).But there are not any information about the method for estimate the stages of dental wear.
-There is no information about the age of individuals. But in the discussion the age category ("juvenis") was introduced (line 170).
- The authors mention about deciduous teeth, but the Dean and Reid method applies only to permanent teeth ( line 144, 149). So it does not really make sense to write about decidous teeth in this context.
- In the paragraph "results" the authors compare the Bucheye Knoll population with the other groups (line 160). This is a mistake, because such a comparison should be transferred to the discussion.
- The authors mention about the weaning stress in the context of LEH. However, we know that the weaning stress must be assess according to isotopic studies (nitrogen), and LEH can be used only as a indirectly be used for this purpose.
- In literature (line 280) Larson - it should be Larsen.
- In literature (line 328) Wood - it is not found in the text.

Validity of the findings

No comment

Additional comments

The material used for the study is very interesting.But the results are quite poorly presented in this paper.

·

Basic reporting

I suggest the authors start the Introduction with hypoplasia (lines 44-58), then give the information on the population studied (lines 32-40) and conclude with the third subsection (59-70). The aims and purpose are clearly stated in the sentence (lines 41-43) ''Here, we infer nonspecific nutritional and developmental stresses via the developmental timing and frequency of linear enamel hypoplasia defects (LEH) in the canines using photogrammetric methods.'', and it would be better if the Introduction finished with it.

Experimental design

No comment

Validity of the findings

The conclusions are connected to the original question investigated, which should be more cleary stated in the Introduction section, as suggested above.

---

## Round 0.2 · accepted · Accept

I think that the vast majority of the issues raised have been efficiently addresses